# Atypical Tongue Abscesses Mimicking Submucosal Malignancies: A Review of the Literature Focusing on Diagnostic Challenges

**DOI:** 10.3390/cancers15245871

**Published:** 2023-12-17

**Authors:** Andrea Rampi, Alberto Tettamanti, Ilaria Bertotto, Lara Valentina Comini, Bright Oworae Howardson, Paolo Luparello, Davide Di Santo, Stefano Bondi

**Affiliations:** 1Otorhinolaryngology Unit, Division of Head and Neck Department, IRCCS San Raffaele Scientific Institute, 20132 Milan, Italy; rampi.andrea@hsr.it (A.R.); tettamanti.alberto@hsr.it (A.T.); howardson.bright@hsr.it (B.O.H.); 2School of Medicine, Vita-Salute San Raffaele University, 20132 Milan, Italy; 3Radiology Unit, Candiolo Cancer Institute, FPO-IRCCS, Candiolo, 10060 Turin, Italy; ilaria.bertotto@ircc.it; 4Otorhinolaryngology Unit, Head and Neck Surgery, Candiolo Cancer Institute, FPO-IRCCS, Candiolo, 10060 Turin, Italy; lara.comini@ircc.it (L.V.C.); paolo.luparello@ircc.it (P.L.); davide.disanto@ircc.it (D.D.S.)

**Keywords:** tongue abscess, tongue malignancy, oral tumors, tongue base

## Abstract

**Simple Summary:**

Even if tongue abscesses are rare, they have very heterogeneous clinical, laboratory and radiological findings. In some cases, their presentation can be particularly nuanced, and they can easily be mistaken for a submucosal malignancy, leading to a dangerous delay in diagnosis. The present paper presents a review of the literature on the possible manifestations (including symptoms, clinical and laboratory findings, imaging) and the management of tongue abscesses in relation to the differential diagnosis of oral tumors.

**Abstract:**

Tongue abscesses are rare conditions that usually follow mucosal disruption due to mechanical trauma or foreign body impaction. They typically manifest abruptly as a rapidly growing, hard mass or swelling in the context of tongue muscles; the patient frequently complains of pain, difficulties in swallowing or speaking, and fever. Nonetheless, the features of its presentation, together with accurate clinical evaluation, blood tests, and appropriate imaging tests, are usually sufficient to easily discern a tongue abscess from a malignancy. However, in rare cases, they may occur with slowly progressing and subtle symptoms, nuanced objective and laboratory findings, and inconclusive radiological evidence, leading to difficult differential diagnosis with submucosal malignancy. Herein, we review the literature, available on Pubmed, Embase, and Scopus, on publications reporting tongue abscesses, with atypical presentation suggesting an oral tumor. Our review confirms that tongue abscesses may manifest as a slowly growing and moderately painful swelling without purulent discharge and minimal mucosal inflammation; in this case, they may constitute an actual diagnostic challenge with potentially severe impact on correct management. Atypical tongue abscesses must therefore be considered in the differential diagnosis of tongue malignancy with submucosal extension, even when other diagnostic elements suggest a neoplasia; in this case, a deep biopsy under general anesthesia is essential for differential diagnosis, and simultaneous drainage of the necrotic and abscessual material may resolve the condition.

## 1. Introduction

A tongue abscess is a rare condition, which is not devoid of potentially life-threatening complications [1]. Its clinical diagnosis can be immediate in the presence of typical signs and symptoms such as significant localized swelling, severe pain, and distinctive radiological and laboratory findings, and prompt surgical and medical treatment is frequently sufficient to prevent further spread of the infection [2,3]. Notwithstanding, tongue abscesses may also manifest with nuanced and atypical features: their clinical presentation can thus vary significantly depending on its promoting factors (such as trauma), localization in muscle tissue, rapidity of the onset of symptoms, and evolution of the disease [4]. Analogously, adequate imaging and laboratory findings may be insufficient to allow for a univocal diagnosis [4,5,6].

For the abovementioned reasons, differential diagnosis can be extremely challenging for the clinician, but the potentially rapid worsening impels its early definition to allow for prompt and appropriate treatment [7].

The recent literature has been reviewed to provide a comprehensive overview of the possible presentations of tongue abscesses and their most appropriate management as a useful tool for any clinician having to face this rare but often challenging condition.

## 2. Materials and Methods

### 2.1. Information Sources and Search

In August 2023, the Pubmed/Medline, Scopus, Embase, and Cochrane databases were searched for publications reporting cases of tongue abscesses. The following medical subject headings (MeSH) were used: “tongue abscess”, “lingual abscess”, and “glossal abscess”. The present study was conducted according to the Declaration of Helsinki.

### 2.2. Exclusion Criteria

The review was restricted to papers written in English and published after 1999; this limit was established to include reports whose setting was more adherent to current clinical practice, since the diagnostic (especially the widespread use of magnetic resonance imaging [MRI]) and therapeutic tools have changed in the last decades. As the focus of the present work was to outline the features of atypically presenting, difficult-to-diagnose abscesses, cases of patients with a relapsing form, an origin from a foreign body inferable from medical history, and infections having origin from a malignancy (cancer abscess) were also excluded. In order to investigate a more homogeneous population, reports on children (age < 18 years) were not included.

### 2.3. Data Synthesis and Descriptive Analysis

After removal of duplicates and a first screening through the abstracts, the full texts of the remaining papers were examined, leading to the inclusion of 28 articles (for a total of 43 patients) meeting the eligibility criteria. From each publication, when available, the following features were extracted: localization within the tongue, symptoms and temporal evolution, clinical presentation, laboratory and radiological findings, microorganisms identified, comorbidities, local predisposing factors (including tabagism), and the medical and/or surgical approach chosen (including the antibiotics administered and the surgical approach and its extension). These findings were then summarized in the present narrative review, with a particular focus on features considered “atypical”, mostly referring to its subtle presentation as well as clinical findings more suggestive of malignancy than a purulent collection (e.g., normal white blood count and inflammatory markers).

## 3. Evidence from the Literature

The review of the literature confirmed that tongue abscesses are an extremely variable condition in terms of symptoms, onset, and radiologic and laboratory findings; furthermore, the clinical presentation and subsequent management seem to be mainly influenced by the localization. In particular, abscesses occurring in the posterior third appear to have distinct features compared to those occurring in the anterior two-thirds, as detailed later.

In the following sections, a descriptive review of these features is provided, while the specific findings are shown in Table 1.

### 3.1. Predisposing Factors

It is worth stressing that patients with an anamnesis positive for a recent acute trauma of the tongue (e.g., fishbone or single bite trauma), clearly suggesting a cause for an abscess, were excluded from the review; on the other hand, cases of chronic trauma, dental surgery, or conditions of chronic oral inflammation, whose differential diagnosis with a malignancy is more challenging, were included.

Specifically considering local factors, several patients were reported as having poor oral hygiene/caries or having recently undergone dental surgery, and many also admitted to active smoking or alcohol abuse [3,6,8,9,10,11]; some authors reported a previous infection of the upper airways [4,12,13]. Interestingly, the patient in the article by Veloo et al. [14] had regular use of corticosteroid inhalers; the role of these agents on superficial oral infections is well known, while there is no evidence of a possible role in deeper soft tissue involvement.

The clinical history also indicated that some patients had immunocompromising features of different severities, including immunosuppressants or immunomodulators, chemotherapy, and diabetes [2,8,15,16]. 

### 3.2. Symptomatology and Clinical Findings

Our review showed odynophagia as the most frequent symptom reported (in all but six patients), but with variable severity, ranging from mild discomfort to the complete impossibility to swallow due to pain. In contrast, fever was associated in a smaller proportion of patients, thus advocating not to exclude a purulent collection in a non-febrile individual. Difficulty in speech/muffled voice was frequently reported [3,17,18,19], while trismus can affect patients with posterior abscesses extending to masticatory muscles [2,3]; referred otalgia was exceptional [11].

At physical examination, a consistent swelling was present in all cases, with some also reporting its extension to submandibular spaces [10], tongue elevation [3], or an anterior protrusion of the jaw [3]. The tongue consistency ranged from a hardened area without a clear firm mass to an indurated lesion with a well-delimited body or a fluctuant softer swelling. Interestingly, in the majority of cases, the overlying mucosa had no clear signs of inflammation; on the other hand, some authors reported the presence of diffuse erythema or exudate [18,20,21].

In most cases, the time interval from the first appearance of symptoms to medical contact was within a week. Nonetheless, a subtle onset was possible, with a time frame up to several weeks [19].

### 3.3. Laboratory Findings

Laboratory tests are an imperative assessment when an abscess is suspected, but it is worth stressing that these can be delayed when a malignancy is the primary suspicion. Notwithstanding, in our review, only rarely did patients have a clear leukocytosis with neutrophilic predominance: in the majority of cases in which it was specified, the white blood cell count remained under 12 × 10^9^/L [2,8,13,15,20,22]. Analogously, only a small increase in inflammatory markers was found in many patients [7,16,20].

From a microbiological point of view, it is interesting to note the wide range of bacteria isolated: different species of Streptococci, Staphylococci, and anaerobes were the most frequent, but in different associations, with many authors simply reporting a mixed oral flora or not able to identify the responsible pathogen [23]. In this regard, the report by Veloo et al. [14] has specific relevance: his group thoroughly studied a case of a lingual abscess using a wide panel of aerobe and anaerobe cultures and molecular techniques (e.g., FISH), with a surprising finding of 12 bacteria isolated from the purulent collection. The same paper (dated 2011) also provided a table with all the recent cases of oral abscesses from the literature with the culture results available: this review was in line with their case, documenting various and often miscellaneous bacterial isolations and some cases in which no microorganism growth was detected. Surprisingly, some authors did not report an attempt at bacterial isolation, but this could be attributable to both logistic difficulties or a negative result that was not specified in the publication [9,14,24]. 

### 3.4. Radiological Findings

The preferred radiological method was contrast-enhanced CT, consistently revealing a hypodense lesion with peripheral enhancement [11]. The reason for its more frequent use can be due to lower costs and wider availability even in peripheral centers and in an urgent setting. It is also worth stressing that the tissue swelling, especially if posterior, may lead to airway flow impairment in a supine position, hindering the feasibility of MRI, which requires a longer acquisition time [12].

Sonography was performed only in four cases [4,6,13,25], revealing a hypoechoic mass with a hyperechoic ring; its use can thus allow the concurrent image-guided aspiration of the purulent collection. Nonetheless, its difficult viability was emphasized, since the significant swelling frequently gives rise to severe pain when the sonograph is applied to the tissue. Furthermore, in the publication by Pallagatti et al. [7], sonography improperly suggested a hemangioma and a drainage was therefore initially avoided for fear of a hemorrhage.

As previously mentioned, multiplanar and multiparametric MRI, despite its high intrinsic tissue contrast resolution, was the method of choice in only a minority of cases [6,26]. 

### 3.5. Management

The correct therapeutic management is obviously subject to prompt and correct diagnosis. Broad-spectrum empirical antibiotic therapy was introduced in all cases after suspicion of an infection: in most reports, it consisted of a highly effective penicillin/cephalosporin and an anaerobe-targeted drug (i.e., clindamycin or metronidazole), with the variable association of vancomycin or amikacin. A systemic corticosteroid was added in some cases to reduce tissue swelling [3,27]. Nonetheless, medical treatment alone was almost invariably insufficient for resolution of the purulent collection and a drainage was deemed necessary [18,28].

Different surgical procedures were performed: for an anterior abscess, both an open-field approach (also in an outpatient setting) and a needle aspiration were used; for a posterior localization, a surgical approach under general anesthesia was preferred, mainly due to greater concerns for airway patency [29]. The surgeon used a transoral or transcervical approach depending on the collection localization [10,27,29]. It is worth mentioning that some patients treated with needle drainage suffered early recurrence, requiring a subsequent open approach [23,27]. 

In two cases of a middle/posterior abscess, the lingual swelling was so significant that it hindered visualization of the vocal cords during nasofibroscopy, a fearsome condition for airway patency [4,17]. Their management was thus different: in one case, a videolaryngoscope was necessary for intubation, but after surgical drainage, a tracheostomy was deemed unnecessary [16]. In the other case, a tracheostomy under local anesthesia was preventively performed to allow for safe ventilation [15]. The concern for the airway patency led to the execution of a tracheostomy in the other four cases (one was described as emergent) [15,22,29]. The particular danger that is intrinsic to a posterior localization is made clear in the report by Schweigert et al. [1]: an unsuccessful intubation was attempted and the patient suffered irreversible cardiac arrest due to hypoxia during the tracheostomy.

**Table 1 cancers-15-05871-t001:** A succinct overview of the studies included for the present review.

First Author	Pt	Symptoms (Days from Onset)	Clinical Findings	Laboratory Findings ^	Imaging/Localization of the Abscess	Comorbidities	Local Predisposing Factors	Atypical Features *^	Management °
Lefler [3]	1	T, O, Sp	To elevation w/subglossal sw.	WBC 18.200	CE CT/BOT	Renal Carcinoma	Periodontitis + tabagism	A	ATB(Clinda + Vanco)
Kettaneh [2]	1	Facial pain (5)	Submandibular sw.; distal To edema	WBC 11.000	CE CT/anterior To	RA	NA	No T; No WBC	ATB(Ampicillin/Sulbactam + Vanco)
Vellin [12]	1	Aphagia (3)	To sw.	WBC 13.000; Gram + cocci and Anaerobes	CE CT/BOT	NA	Previous pharyngitis + tobacco	Light WBC	FNA(GA) +ATB (Cef + Metro)
Solomon [30]	1	T, O, Dy (7)	To firm mass w/limited mobility	NA	CE CT	NA	NA	Long Sy	Surgery + ATB
Veloo [14]	1	Pain, Sp (7)	To tender sw.	WBC 12.000; 12 bacteria species isolated	CE CT/central To	Asthma in tx w/budenoside	NA	No WBC; Long Sy	FNA + Surgery (drainage) + ATB (Amoxi/Clav)
Antoniades [8]	3	1. Dys, Tr, Dy; 2. O, Dy, Sp, Tr; 3. T(38.9), Dy, Sp	1. To sw. w/indurated margins; 2. To sw. w/debris; 3. Erythematous sw.	1. WBC 4.600, S. Faecalis; 2. WBC 14.300, Strep + anaerobes; 3. WBC 15.000, Bacteroides species	1;2;3:NA	1. NA; 2. Chemotherapy for chronic leukaemia; 3. DMII	1. Tooth extraction, alcohol abuse; 2. Poor oral Hy; 3. Oral trauma	No WBC	1. FNA + ATB (Cefuroxime + Amikacin + Metro); 2. FNA + ATB (Ceforanide + Amikacin + Metro); 3. S urgery + ATB(Ticarcillin/Clav; + Amikacin)
Potigailo [17]	1	O, Dy, Sp	Cervical Lymp; BOT and epiglottic sw.	NA	CE CT/BOT	NA	NA	A	FNA + ATB
Little [27]	1	O (5)	Immobile To, Bilat. sublingual sw.	WBC 17.600, S. Anginosus+ Fusobacterium nucleate	CE CT/BOT	NA	NA	No response to double ATB	FNA + Surgery(transcervical) + ATB(Ampicillin/Sulbactam + Vanco)
Harrington [23]	1	O, Dy (5)	To and FOM sw.	WBC 21.500; S. Intermedius	CE CT/anterior To	NA	Recent tooth abscess	A	FNA + Surgery + ATB(Clindamycin)
Kikidis [9]	1	Sp (7)	To sw.	NA	CE CT	NA	Poor oral Hy.	Long Sy	FNA + Surgery + ATB(Amoxi/clav + Metro)
Ozgur [13]	1	O, Dy (30)	BOT sw.	WBC 6.940, CRP 51, coagulase-negative Staph.	CE CT/BOT	NA	Recent tx for pharyngitis	Long Sy; Weight loss; No WBC	FNA + ATB(Cef)
Akin [15]	3	1. Dy, O (2); 2. O, Dy, Dys (4); 3.O, Dy	1. Painful To sw; 2. Painful To sw; 3. Multilobed abscess	1. WBC 16.600, CRP 9.94, Group-D beta-haemolytic Strep; 2. WBC 10.900, CRP 26.18; 3.WBC 5.600, CRP 207, mixed oral flora	1. CE CT/middle To; 2. MRI/middle To; 3. CE CT	1. NA; 2. HT; 3. Acute lymphoblastic leukaemia under Chemo	1. Previous dental intervention; 2. NA; 3. Poor oral Hy.	1;2 No T 2;3 No WBC	1. Surgery(GA) + ATB(Cef + Metro); 2. Surgery(GA) + tracheostomy + ATB(Piperacillin/Tazobactam + Teicoplanin)
Haydar [24]	1	O, Dy, Sp (5)	To sw. w/hyperemia	WBC 15.700, CRP 50	CE MRI/anterior To	NA	NA	No T	Surgery + ATB (Cef + Clinda)
Mesolella [4]	1	Sp, Dy, O (15)	To sw., on nasofibroscopy no visualization of vocal cords	WBC elevated, Staph. species and anaerobes	CE CT + MRI/BOT	NA	Previous tonsillitis	Long Sy	Surgery(GA) + ATB(Cef)
Saro-Buendía [18]	2	1. T, O, Sp; 2. O, Sp (3)	1. BOT tumefaction w/erythema; 2. Vallecula tumefaction w/erythema	1. WBC elevated, CPR 92.8; mixed flora w/*Strep* spp. and *Fusobacterium* spp.; 2. WBC elevated; CRP 30.4; mixed flora	1. CE CT/BOT; 2. CE CT/vallecula and BOT	NA	NA	A	1. Surgery + ATB(Cef + Clinda); 2. Surgery + ATB
Riccardi [20]	1	Dy, Sp (1)	To sw. w/papule on the dorsum	CRP 6.9	US/right hemiTo	NA	NA	No T; slight CRP	Surgery
Binar [10]	1	O, Dys, Sp (3)	Neck sw., BOT sw., on nasofibroscopy no visualization of vocal cords	Hemophilus influenza type b	CE CT/BOT	NA	Poor oral Hy	No WBC	Surgery (transcervical) + ATB(Ampicillin/Sulbactam)
Bekele [19]	1	O, Dy, Sp (40)	To sw., pus oozing from the sw	Normal range; Gram + cocci	NA	NA	Poor oral Hy; previous tooth extraction	Long Sy; No WBC	Surgery + ATB(Amoxi/Clav. + Metro)
Varghese [22]	1	Dy, Dys (5)	Anterior 2/3 To sw. occluding oropharynx	WBC 11.500	CE CT/anterior 2/3 To abscess	NA	Recent tooth extraction	No WBC	Surgery(GA) + tracheostomy + ATB
Burnham [16]	1	O (7)	To sw	WBC 11.340; PCR 24; Gram + and Gram-cocci	CE CT/midline and BOT	DMII	NA	Long Sy; No WBC	FNA + Surgery(GA) + (Amoxi-Clav)
Eviatar [21]	1	T, O, Dy (4)	BOT sw., white exudate at palatine tonsils	WBC 10.800, Prevotella	CE CT/BOT	NA	NA	No WBC	FNA + ATB (Amoxi-Clav)
Mesfin [25]	1	T, Sp, O	Tender To, erythema and aphthous ulcers	NA	NA	NA	Khat chewer, poor oral Hy.	A	Surgery(GA) + ATB Cef + Metro)
Pallagatti [7]	1	O, Dy (4)	Anterior 2/3 To sw.	Normal range, Fusobacterium nucleatum, Prevotella and *Strep* spp.	US/anterior 2/3 To	NA	NA	Normal range	FNA + ATB(Amoxi + cloxacillin)
Stofferahn [31]	1	O, Dy, Sp (5)	To sw., erythema	WBC 17.400, Group B strep, oral flora	CE CT/Posterior 1/3 To	HT	NA	A	FNA + ATB(Ampicillin/Sulbactam + Linezolid)
Srivanitchapoom [29]	6	1. T, O, Dy, Dys; 2. O; 3. O, Dy; 4.O, Dy, Dys; 5. Otalgia, O; 6.O	1. BOT Marked sw.; 2. Antero-lateral To sw; 3. FOM and BOT Sw.; 4. FOM Sw.; 5.BOT Sw.; 6. Antero.midline To mass	1. WBC 9300; S. Viridans; 2. WBC 4500; 3. WBC 14.500, Acitenobacter lwoffii; 4. WBC 12.100, *Strep* spp.; 5.WBC 5500; S. Viridans; 6. WBC 5800	1;3;4;5: CE CT 2;6: None	1;3;5;6: None; 2;4:DM + HT;	1;2: Poor oral Hy; 3;4;6:NA; 5. Poor oral hy + Thyroglossal duct cyst	2;3;4;5;6 No T 1;2;4;5;6 No WBC	1. Surgery + tracheostomy + ATB(Amoxi/Clav + Cef); 2. Surgery(LA) + ATB(Amoxi/Clav+ Cef); 3.Surgery(AG) + ATB therapy (Clinda; Cef); 4.Surgery(GA) + tracheostomy + ATB(clindamycin; Cef); 5. Surgery(GA) + ATB(Amoxi/Clav + Cef); 6. Surgery(GA) + ATB(Amoxi/Clav)
Balatsouras [11]	4	1. Otalgia, O; 2. O, Sp; 3. Sp, O; 4. O, otalgia	1. Sw. BOT 2. sw middle To and BOT, on nasopharyngoscopy edema of the epiglottis and left aryepiglottic fold 3. sw. of the middle portion of To 4. abscess at BOT and edema of epiglottis	1. Strep.	1;2;3: CE CT 4. None	1; 4 DMII 2; 3 None	1; 2; 3; Poor oral hy. 4. None	A	1. FN +ATB(Penicillin + Gentamycin + Metro); 2. FNA + ATB(Penicillin+ Gentamycin + Metro); 3. FNA + ATB(Amoxi/Clav); 4. FNA + ATB(Amoxi/Clav)
Schweigert [1]	1	Sp; Dys; Dy	To sw.	NA	NA	CAD	NA	A	Emergent tracheotomy (patient deceased)
Ozturk [6]	7	1. T, O; 2. T, O; 3. O; 4. O, Dys; 5. O; 6. Dys; 7. O	1; 2; 3; 4; 5; 6; 7 To mass	1; 2 Leukocytosis; anaerobic bacteria; 3. Normal range, S. Viridans; 4; 5; 6; 7 Normal range	1; 2; 3; 4; 5; 6; 7 CE MRI	1. alcohol abuse	1; 2; 3 Poor oral Hy	3;4;5;6;7 Normal range	1; 2; 3; 4: Surgery(GA) 5; 6; 7: FNA

° When a cervical approach was necessary, it is specified, otherwise surgery refers to a transoral approach ^ Normal range specifies WBC ≤ 12 × 10^9^/L w/no inflammatory markers elevated. * A refers to an anamnesis not suggestive for an abscess or elements supporting a cancer, as neoplasms, Khat chewing, etc.; long symptoms are defined with a duration ≥ 7 days; no WBC when ≤ 12 × 10^9^/L. Abbreviations: ATB: Antibiotic; Bilat: Bilateral; BOT: base of the tongue; CAD: coronary artery disease; CE: contrast-enhanced; Cef: Ceftriaxone; CT: Computed tomography scan; CRP: C- reactive protein (mg/L); Clinda: Clindamycin; Dy: Dysphagia; Dys: Dyspnea; DMII: Diabetes type II; FNA: Fine needle Aspiration; FOM: Floor of the mouth; GA: General Anaesthesia; Hy: Hygiene; HT: Hypertension; Lym: Lymphadenopathy; LA: Local anaesthesia; Metro: Metronidazole; NA: not applicable or not reported; O: odynophagia; pt: patient(s); RA: Rheumatoid Arthritis; Sp: difficulty in speech; Sy: symptoms; Sw: swelling; T: fever; To: Tongue; TR: trismus; Tx: treatment; Vanco: vancomycin; WBC: white blood cells (×10^9^/L); w/: with.

## 4. Discussion

Lingual abscesses are a rare condition, typically affecting young to middle-aged adults [29]. Their uncommon occurrence has been attributed to the efficient defensive properties of the tongue: the immunological properties of saliva can promptly neutralize bacteria; the thick keratinized mucosa, together with a cleansing action of saliva via continuous tongue movement, further hinder the penetration of microorganisms, where the rich vascular supply and lymphatic drainage of the muscles allow for their eventual early eradication [9].

An abscess in the context of the tongue is therefore frequently associated with a mechanical factor that can disrupt these barriers, and the diagnosis is therefore immediate. Various other conditions, often more difficult to identify, have been associated, particularly with posterior abscesses such as infected thyroglossal duct cyst remnants, lingual tonsillitis, and infections spreading from lower molar teeth [13,21]. Our review shows that a purulent collection can also originate in the absence of these conditions in the patient’s medical history: an unrecognized microtrauma and systemic conditions hindering an immunological response are suspected to play a significant role [20,31].

Given its rarity, correct diagnosis can be particularly challenging, as highlighted in the present review. In fact, the clinical presentation can be particularly subtle, with a slowly growing hardened/fluctuant lesion leading to discomfort, odynophagia, weight loss, and occasional fever. The differential diagnosis includes inflammatory and infectious diseases, such as actinomycosis, granulomatous lesions of different origins or IgG4-related disease, and congenital lesions or tumoral masses [5,32,33,34]. Focusing on the latter group, submucosal spread can be present in primary tumors of mesenchymal (e.g., leiomyoma, lipoma, and rhabdomyosarcoma) or salivary origin (particularly pleomorphic adenoma, mucoepidermoid, and adenoid cystic carcinoma), but also squamous cell carcinoma, which almost invariably affects the epithelial lining, has been seldomly reported as a submucosal hardened lesion [35,36,37,38,39,40,41]. The tongue can also be the site of lymphoproliferative disease (primarily B-cell, non-Hodgkin lymphoma) or distant metastasis, mainly developing from lung and renal carcinoma or cutaneous melanoma [5,38,42,43,44,45]. Notwithstanding, a tongue abscess can be more easily mistaken for a malignancy in the event of unusually long-lasting symptoms and this error can lead the clinician to avoid fundamental evaluations such as laboratory tests and immediate imaging.

In this regard, it is worth stressing that the symptoms are primarily related to the localization of the disease, with posterior abscesses determining a swelling that potentially leads to emergent airway impairment and greater difficulties in swallowing [17]. Mesolella et al. [4] furthermore differentiated between superficial/submucosal and deep abscesses, suggesting that the former have a more subtle presentation, as specifically investigated herein, and the latter to induce a more severe inflammatory process with systemic involvement (including fever, leukocytosis, and general weakness). This distinction seems to be confirmed by our review, despite the limited number of patients.

The clinical presentation may thereby confuse the clinician, and other diagnostic tools can also be inconclusive; in this regard, many patients show a normal white blood cell count and temperature (or only slightly altered), without an increase in inflammatory markers [7,16,20].

Any clinician having to face a patient with these difficult features must therefore consider a subtle-presenting abscess in the differential diagnosis, and immediate evaluation by an otorhinolaryngologist, together with a course of broad-spectrum empiric antibiotics and proper imaging, is advised. While appropriate medical therapy can avoid further extension of the disease, and even treat more localized infections such as cellulitis (which is often a differential diagnosis), imaging is in fact fundamental, both for the formulation of diagnostic hypotheses and to direct therapeutic management [22]. CT can thus be insufficient for differential diagnosis and many artifacts (e.g., metal dental materials) can hinder proper visualization of the disease [6]. MRI can be preferred for its high tissue contrast resolution, with CT still having a complementary role in determining the extension of the inflammatory process to bone structures [6]. In some cases, imaging findings are unequivocal and can guide the surgical maneuver: in MRI, the evidence of a T1-hypointense, T2 hyperintense lesion with a peripheral T1-hyperintense, T2-hypointense rim, enhanced after gadolinium injection, is highly suggestive of a lingual abscess [6]; perilesional edema of various extensions is a frequent ancillary finding [6,26]. Otherwise, it can also be detected as a diffuse or peripheral-enhancing mass-like finding [6]. In some cases, the radiologic appearance can thus be inconclusive in the differential diagnosis between an abscess and a submucosal malignancy (Figure 1) [5,42]. In particular, high grade tumors with intralesional necrosis and poorly-defined borders are the most difficult to distinguish, although some features, such as the paramagnetic property of melanin at MRI in metastatic melanoma, can be particularly informative [5,38,42].

A paramount finding of the present review is that a low threshold for a direct, prompt approach is advisable in uncertain cases where all other diagnostic tools are insufficient: it is, in fact, worth stressing that the limits to the penetration of antibiotic therapy render a surgical approach fundamental for resolution of an abscess; in the suspicion of a malignancy, it is likewise diagnostic, providing a direct vision and the possibility for histologic/cytologic evaluation of tissue [8]. Needle aspiration and surgical procedures have both been described, and their choice is dependent on the localization, risks, and familiarity of the clinician with these techniques. If a first unsuccessful attempt is performed in an outpatient setting, a second one in the surgical theater under general anesthesia appears rational to maximize the possibility of success [27]. Purulent evacuation can be performed via a transoral or transcervical/combined approach, mainly depending on the localization of the disease and its extension to neighboring structures. In this context, a temporary tracheostomy represents a crucial decision, which is often staunchly opposed by the patient. Once more, the choice is strictly dependent on the specific case, but the surgeon must be aware of the risk for airway patency, especially in the event of a posterior abscess: post-surgical edema or an intramuscular hemorrhage can determine an acute obstruction. In consideration of the limited long-term morbidity of the procedure, in contrast to a potential life-saving outcome, a low threshold for its execution is advised. When there is concern about the feasibility of tracheal intubation, a tracheostomy under local anesthesia avoids the risks associated with the induction of narcosis, as suggested by the fatal outcome in the report by Schweigert et al. [1].

A culture and antibiogram from the purulent material obtained is always appropriate and may sustain the tapering of the previously prescribed therapy or its modification when the microorganism is resistant. A negative culture is frequently reported, as much as the growth of a saprophytic flora or multiple pathogens, suggesting a primary role of an impairment in mucosal and immunological defense rather than an intrinsic aggressiveness of the bacteria involved [14]. This aspect, on one hand, reveals the possible difficulties in the introduction of appropriate, targeted antimicrobial therapy; on the other, it supports the need for early mechanical removal of the purulent material.

The strict methodology applied in the present review represents its main strength, and the large number of publications evaluated guarantees the inclusion of a broad spectrum of different cases in terms of symptoms, localization, and management. On the other hand, this heterogeneity represents a limit for the generalization of conclusions; additionally, a publication bias in favor of the most peculiar conditions may have overestimated the atypical presentations.

## 5. Conclusions and Future Directions

Herein, we provide an overview of the main diagnostic and therapeutic difficulties of atypical lingual abscesses in relation to a possible presentation resembling a submucosal malignancy. The diagnostic challenge offered by a case with a subtle presentation must be faced via the integration of clinical, laboratory, and imaging findings, which can be nonetheless inconclusive. A needle aspiration and/or an open surgical approach are therefore advocated in these rare cases to provide a definitive diagnosis and simultaneously represent an unavoidable therapeutic tool; resorting to them should therefore not be delayed. In the future, better definition of the main features and proper management on the basis of the abscess localization (anterior/posterior, superficial/deep) is advisable.

## Figures and Tables

**Figure 1 cancers-15-05871-f001:**
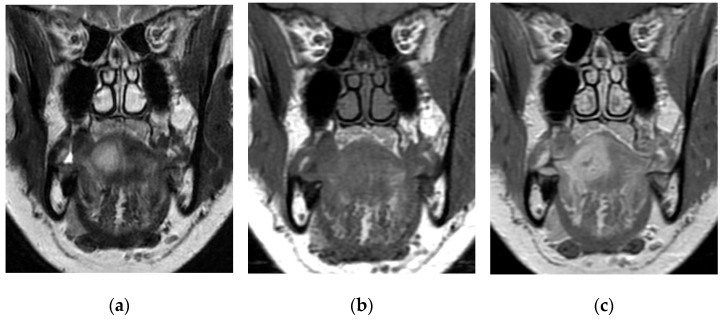
MRI on the coronal plane of an abscess in a 55-year-old woman presenting at our hospital for a slowly growing painful mass developing over a 2-month interval, with normal white blood cell count, initially suspected as a submucosal malignancy. A mass embedded in the intrinsic muscles of the right hemitongue, reaching the midline, is evident. The lesion presents hyperintensity in the T2-weighted image with a low signal peripheral rim (**a**) and intermediate signal intensity in the T1-weighted image (**b**). In the T1-weighted image after gadolinium injection, intense contrast enhancement is observed, with the exception of a limited central portion of the mass (**c**). The imaging is suggestive of an abscess, but a submucosal neoplasm could not be ruled out.

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
