# Peer review of "Atypical Tongue Abscesses Mimicking Submucosal Malignancies: A Review of the Literature Focusing on Diagnostic Challenges"

_cancers, 2023, doi:10.3390/cancers15245871_

Round 1
Reviewer 1 Report
Comments and Suggestions for Authors
1.The article is based on a very interesting premise.However,the title is misleading since the authors have not described in detail the differential diagnosis of tongue abscesses with submucosal malignancies as suggested in the title.
2.There are quite a few grammatical errors present throughout the manuscript which has been duly highlighted in the reviewed manuscript
3.Quite a few sentences are incoherent(highlighted in the manuscript) and needs to rephrased for a better understanding

Comments on the Quality of English Language1.The article is based on a very interesting premise.However,the title is misleading since the authors have not described in detail the differential diagnosis of tongue abscesses with submucosal malignancies as suggested in the title.
2.There are quite a few grammatical errors present throughout the manuscript which has been duly highlighted in the reviewed manuscript
3.Quite a few sentences are incoherent(highlighted in the manuscript) and needs to rephrased for a better understanding
Author Response
TO THE EDITOR:
We deeply appreciated the suggestions made by the reviewers and you on our manuscript entitled “Atypical tongue abscesses mimicking submucosal malignancies: our experience and a review of the literature”. We modified the manuscript according to the comments, and the specific answers to any comments are listed below. In particular, the case presentation was removed, while the discussion was implemented, mainly to address the differential diagnosis. The title of the paper has been therefore changed to “Atypical tongue abscesses mimicking submucosal malignancies: a review of the literature focusing on diagnostic challenges”.
In addition, a linguistic revision was performed by a mother-tongue reviewer, as suggested, and the layout of the table was improved: in particular, the “Pt” column was tightened in favor of the “first author” column.
We hope that the changes made and the following explanations will clarify any doubts expressed by the reviewers and that you will find the manuscript worthy of publication.
Kind Regards,
Stefano Bondi
Reviewer #1, # 2; #3 and #4: we thank you for your review and the precious suggestions you have given. According to these indications, the manuscript has been modified and implemented.
Reviewer #1
1.The article is based on a very interesting premise. However, the title is misleading since the authors have not described in detail the differential diagnosis of tongue abscesses with submucosal malignancies as suggested in the title.
We thank you for your appreciation and your valid insight. We have widened, in the discussion section, the topic of differential diagnosis, displaying the wide panel of possible lesions (see in particular lines 217-226). We have also rephrased some parts to emphasize that in some cases there is no tool reliable enough to discern these conditions, and a direct approach is therefore advisable to drain the abscess or, otherwise, to obtain histological confirmation of a neoplastic lesion (see lines 252-276). We hope that these changes could better convey some of the main findings of the present review.
2. There are quite a few grammatical errors present throughout the manuscript which has been duly highlighted in the reviewed manuscript 3. Quite a few sentences are incoherent(highlighted in the manuscript) and needs to rephrased for a better understanding
Your review has been particularly accurate and valuable in addressing the incorrect or less coherent parts in our paper, and the pdf with your indication was really helpful. We have modified the article accordingly. In addition, a linguistic revision has been performed by a mother-tongue reviewer.

Reviewer 2 Report
Comments and Suggestions for Authors
Well presented case report with clinical interest
Author Response
TO THE EDITOR:
We deeply appreciated the suggestions made by the reviewers and you on our manuscript entitled “Atypical tongue abscesses mimicking submucosal malignancies: our experience and a review of the literature”. We modified the manuscript according to the comments, and the specific answers to any comments are listed below. In particular, the case presentation was removed, while the discussion was implemented, mainly to address the differential diagnosis. The title of the paper has been therefore changed to “Atypical tongue abscesses mimicking submucosal malignancies: a review of the literature focusing on diagnostic challenges”.
In addition, a linguistic revision was performed by a mother-tongue reviewer, as suggested, and the layout of the table was improved: in particular, the “Pt” column was tightened in favor of the “first author” column.
We hope that the changes made and the following explanations will clarify any doubts expressed by the reviewers and that you will find the manuscript worthy of publication.
Kind Regards,
Stefano Bondi
Reviewer #2
Well presented case report with clinical interest
We thank you for appreciating our work.

Reviewer 3 Report
Comments and Suggestions for Authors
The submitted manuscript describes the case presentation of tongue abscess with clinically-diagnostic difficulty with a literature review. The reviewer considered that the topic is interesting but does not fall within the scope of Cancers.
Comments
1. The definition of “atypical tongue abscess” was obscure.
2. Thus, the importance of differential diagnosis from submucosal malignancy, such as metastatic lesions or salivary gland tumors predominantly showing submucosal patterns, is not described well.
3. In the case presentation, the intraoral and biopsy samples’ histological images and description were absent.
Comments on the Quality of English Language
The English writing was fine.
Author Response
TO THE EDITOR:
We deeply appreciated the suggestions made by the reviewers and you on our manuscript entitled “Atypical tongue abscesses mimicking submucosal malignancies: our experience and a review of the literature”. We modified the manuscript according to the comments, and the specific answers to any comments are listed below. In particular, the case presentation was removed, while the discussion was implemented, mainly to address the differential diagnosis. The title of the paper has been therefore changed to “Atypical tongue abscesses mimicking submucosal malignancies: a review of the literature focusing on diagnostic challenges”.
In addition, a linguistic revision was performed by a mother-tongue reviewer, as suggested, and the layout of the table was improved: in particular, the “Pt” column was tightened in favor of the “first author” column.
We hope that the changes made and the following explanations will clarify any doubts expressed by the reviewers and that you will find the manuscript worthy of publication.
Kind Regards,
Stefano Bondi
Reviewer #1, # 2; #3 and #4: we thank you for your review and the precious suggestions you have given. According to these indications, the manuscript has been modified and implemented.
Reviewer #3
The submitted manuscript describes the case presentation of tongue abscess with clinically-diagnostic difficulty with a literature review. The reviewer considered that the topic is interesting but does not fall within the scope of Cancers.
We thank you for your evaluation. Our aim was to display a difficult diagnostic challenge, especially for clinicians who are used to managing Head and Neck malignancies and may not consider rare conditions (like tongue atypical abscesses) in the differential diagnosis, and this is the reason we subumitted this paper to Cancers.
Comments
- The definition of “atypical tongue abscess” was obscure.
We thank you for your observation. Actually, the definition included all the cases with atypical features (such as no fever, no leukocytosis, subtle presentation, nuanced symptoms), but it was not adequately specified. We added it in the material section (see lines 81-84).
- Thus, the importance of differential diagnosis from submucosal malignancy, such as metastatic
lesions or salivary gland tumors predominantly showing submucosal patterns, is not described well.
Your insight is appropriate, as also suggested by another reviewer. We have widened, in the discussion section, the topic of differential diagnosis, displaying the wide panel of possible lesions (see in particular lines 217-226). We have also rephrased some parts to emphasize that in some cases there is no tool reliable enough to discern these conditions, and a direct approach is therefore advisable to drain the abscess or, otherwise, to obtain histological confirmation of a neoplastic lesion (see lines 252-276). We hope that these changes could better convey some of the main findings of the present review.
- In the case presentation, the intraoral and biopsy samples’ histological images and description were absent.
We apologize for this. Nonetheless, the academic author has suggested removing our case presentation to adapt to Cancers’ guidelines, and so all this section is absent in this resubmission.
- The English writing was fine.
We thank you for your appreciation. Following the suggestions of another reviewer, a linguistic revision has been nonetheless performed by a mother-tongue reviewer.

Reviewer 4 Report
Comments and Suggestions for Authors
Dear Authors,
I don't have any specific comments or suggestions. The writing is clear and organized, making it easy for the reader. References are well-chosen. Only what can I say that this case report with a literature review is not of great scientific importance, but it is of importance for everyday clinical practice.
Best regards
Author Response
TO THE EDITOR:
We deeply appreciated the suggestions made by the reviewers and you on our manuscript entitled “Atypical tongue abscesses mimicking submucosal malignancies: our experience and a review of the literature”. We modified the manuscript according to the comments, and the specific answers to any comments are listed below. In particular, the case presentation was removed, while the discussion was implemented, mainly to address the differential diagnosis. The title of the paper has been therefore changed to “Atypical tongue abscesses mimicking submucosal malignancies: a review of the literature focusing on diagnostic challenges”.
In addition, a linguistic revision was performed by a mother-tongue reviewer, as suggested, and the layout of the table was improved: in particular, the “Pt” column was tightened in favor of the “first author” column.
We hope that the changes made and the following explanations will clarify any doubts expressed by the reviewers and that you will find the manuscript worthy of publication.
Kind Regards,
Stefano Bondi
Reviewer #1, # 2; #3 and #4: we thank you for your review and the precious suggestions you have given. According to these indications, the manuscript has been modified and implemented.
Reviewer #4
Dear Authors,
I don't have any specific comments or suggestions. The writing is clear and organized, making it easy for the reader. References are well-chosen. Only what can I say that this case report with a literature review is not of great scientific importance, but it is of importance for everyday clinical practice.
Best regards
We thank you for your revision and for your appreciation. As you suggest, our paper provides a more pragmatic approach useful for everyday clinical practice, but, in our opinion, this does not necessarily imply it lacks scientific importance.

Round 2
Reviewer 1 Report
Comments and Suggestions for Authors
The queries have been satisfactorily rectified by the authors and the manuscript is modified satisfactorily as per the observations
Author Response
Reviewer #1
The queries have been satisfactorily rectified by the authors and the manuscript is modified satisfactorily as per the observations
We thank you for appreciating our effort in the improvement of the present work.

Reviewer 3 Report
Comments and Suggestions for Authors
The authors describe the rarity of tongue abscesses and their subtle symptoms in the modified manuscript. The reviewer considered that the topic is interesting but recommended further improve for the manuscript.
Comments
1. The author described the definition of “atypical tongue abscess” as following the reviewer’s comment. I understood the definition of “atypical” was that the reviewed case met at least one or more described criteria (such as no fever, no leukocytosis, subtle presentation, and nuanced symptoms). However, I think the definition was somewhat subjective. The authors listed “atypical tongue abscesses” that met their criteria of “atypical” in the table. It would be desirable to specify which points were “atypical” in the table of reviewed cases. I apologize for not having noted this in the initial review, as the definition of "atypical tongue abscess" was not clearly stated.
Author Response
TO THE EDITOR:
We deeply appreciated the suggestion made by the reviewer #3 concerning the atypical features of the cases. We added, as suggested, an additional column in the table to better display the atypical features of our cases.
We hope that the changes made will clarify the doubts expressed and that you will find the manuscript worthy of publication.
Kind Regards,
Stefano Bondi
Reviewer #3
The author described the definition of “atypical tongue abscess” as following the reviewer’s comment. I understood the definition of “atypical” was that the reviewed case met at least one or more described criteria (such as no fever, no leukocytosis, subtle presentation, and nuanced symptoms). However, I think the definition was somewhat subjective. The authors listed “atypical tongue abscesses” that met their criteria of “atypical” in the table. It would be desirable to specify which points were “atypical” in the table of reviewed cases. I apologize for not having noted this in the initial review, as the definition of "atypical tongue abscess" was not clearly stated.
We agree that the definition of atypical presentation may be somewhat subjective, but our aim was to include any case which was not explicitly suggestive of an abscess (for example for patients with relapsing forms, or purulent discharge). ENT surgeons must in fact be aware to detect, between the frequent cases of tongue malignancies, the rare and multifaceted lingual abscesses. For this reason, we included patients with long lasting symptoms, without increase in white blood cell count or fever (as in the majority of the cases) but also, more generally, patients with an anamnesis which could support a neoplasm, as no history of previous tongue trauma in a patient with known neoplasia in a different site or a Khat chewer (only a few cases), as displayed in the material and methods section (in particular the subsection “exclusion criteria”). Following your precious suggestion, we added a column in the table specifically addressing the atypical features for each case, and we green-highlighted the changes.
